# Genome Mining and Genetic Manipulation Reveal New Isofuranonaphthoquinones in *Nocardia* Species

**DOI:** 10.3390/ijms25168847

**Published:** 2024-08-14

**Authors:** Purna Bahadur Poudel, Dipesh Dhakal, Rubin Thapa Magar, Niranjan Parajuli, Jae Kyung Sohng

**Affiliations:** 1Department of Life Science and Biochemical Engineering, Institute of Biomolecule Reconstruction (iBR), Sun Moon University, Asan 31460, Republic of Korea; pbspoudel@gmail.com (P.B.P.); medipesh@gmail.com (D.D.); magarrubin@gmail.com (R.T.M.); niranjan.parajuli@cdc.tu.edu.np (N.P.); 2Central Department of Chemistry, Tribhuvan University, Kirtipur, Kathmandu 44618, Nepal; 3Department of Pharmaceutical Engineering and Biotechnology, Sun Moon University, Asan 31460, Republic of Korea

**Keywords:** anticancer, antimicrobial, CRISPR-Cas9, SARP regulator, secondary metabolites

## Abstract

The identification of specialized metabolites isolated from microorganisms is urgently needed to determine their roles in treating cancer and controlling multidrug-resistant pathogens. Naphthoquinones act as anticancer agents in various types of cancers, but some toxicity indicators have been limited in their appropriate application. In this context, new isofuranonaphthoquinones (ifnq) that are less toxic to humans could be promising lead compounds for developing anticancer drugs. The aim of this study is to identify and characterize novel furanonaphthoquinones (fnqs) from *Nocardia* sp. CS682 and to evaluate their potential therapeutic applications. Analysis of the genome of *Nocardia* sp. CS682 revealed the presence of a furanonaphthoquinone (*fnq*) gene cluster, which displays a similar genetic organization and high nucleotide sequence identity to the *ifnq* gene cluster from *Streptomyces* sp. RI-77, a producer of the naphthoquinones JBIR-76 and JBIR-77. In this study, the overexpression of the *Streptomyces* antibiotic regulatory protein (SARP) in *Nocardia* sp. CS682DR (nargenicin gene-deleted mutant) explicitly produced new fnqs, namely, NOC-IBR1 and NOC-IBR2. Subsequently, the role of the SARP regulator was confirmed by gene inactivation using CRISPR-Cas9 and complementation studies. Furthermore, antioxidant, antimicrobial, and cytotoxicity assays were performed for the isolated compounds, and it was found that NOC-IBR2 exhibited superior activities to NOC-IBR1. In addition, a flexible methyltransferase substrate, ThnM3, was found to be involved in terminal methylation of NOC-IBR1, which was confirmed by in vitro enzyme assays. Thus, this study supports the importance of genome mining and genome editing approaches for exploring new specialized metabolites in a rare actinomycete called *Nocardia*.

## 1. Introduction

Natural products play an important role in drug discovery, especially in the treatment of various types of cancers and infectious diseases [1,2,3]. Quinones are a major class of compounds in the fight against cancers because several of their derivatives have been used as anticancer agents [4,5]. However, some toxicity indicators have reduced the pace of global drug development programs. In this regard, new fnqs may serve as lead compounds for the development of anticancer drugs. These compounds are a group of natural compounds distinguished by their unique tricyclic naphtha [2,3-c] furan ring structure, which can have different substituents on rings A, B, or C; these compounds are mostly identified in plants, fungi, and actinobacteria such as *Streptomyces* sp. CB01883. Ifnqs have shown cytotoxic, antiplasmodial, antioxidant, antibacterial, and siderophore properties [6,7,8,9].

In recent years, several specialized metabolites have been discovered through genome mining, which links genetic information with secondary metabolites produced [10]. By analyzing the genome databases of diverse actinobacteria, researchers have identified a surprising number of “silent” or “cryptic” biosynthetic gene clusters (BGCs). These clusters are good sources for the discovery of novel and previously unknown metabolites [11]. There are several ways to activate such silent BGC to extract new metabolites, such as RNA polymerase mutation, elicitor application, heterologous expression, and overexpression of global or pathway-specific regulators [12,13].

Silent BGCs can be induced to produce potentially novel compounds by altering transcriptional regulators in proximity to BGCs. For example, the overexpression of positive regulators and the inactivation of negative regulators [14] and/or the tuning of the expression of regulators using *bldA* tRNA [15] have been utilized in various actinomycetes. In the microbial community, *Streptomyces* antibiotic regulatory protein (SARP) regulators serve as pathway-specific activators of secondary metabolite production [16]. In particular, the N-termini of SARP activators have a winged helix-turning helix motif, which is shared by the OmpR protein family. Furthermore, heptameric repeats found in the promoter regions of genes regulated by SARPs can be recognized by at least some parts of SARPs [16,17]. Overexpression of the SARP regulator resulted in the activation of some cryptic BGCs and/or enhanced the production of various secondary metabolites [16,18]. Recent advancements in different omics techniques, along with the availability of robust genetic engineering techniques such as CRISPR-Cas9, have further facilitated genome mining.

*Nocardia* sp. CS682 is a well-known producer of nargenicin (ngn) A1, which exhibits potent antibacterial activity against a variety of Gram-positive bacteria, including methicillin-resistant *Staphylococcus aureus* (MRSA). To produce new secondary metabolites, we engineered a strain, *Nocardia* sp. CS682DR, by deleting its entire *ngn* BGC. This led to the characterization of a novel tetrahydroxynaphthalene (THN) derivative, IBR1-4 [19]. Bioinformatics analysis revealed the presence of a SARP regulator between the *ngn* BGC and an unidentified type II PKS. This study aimed to identify and characterize new fnqs with potential therapeutic applications, particularly focusing on their anticancer and antibacterial properties and reducing toxicity. To achieve this goal, we overexpressed the SARP regulator in *Nocardia* sp. CS682DR, resulting in the production of new *ifnq*, 6-methoxy-7,8-dihydroxy-1-methylnaphtho [2,3-c] furan-9-one (NOC-IBR1) and 6,7-dimethoxy-8-hydroxy-1-methylnaphtho [2,3-c] furan-9-one (NOC-IBR2). Furthermore, we performed gene inactivation using CRISPR-Cas9 to confirm the involvement of the regulator in the biosynthesis of NOC-IBR1/2. Additionally, evaluation of their bioactivities revealed that NOC-IBR2 exhibited significant anti-*staphylococcal*, anticancer, and antioxidant activities. Remarkably, it was shown that the type II PKS pathway interacts with ThnM3, a methyltransferase located inside the THN BCG. Through this connection, NOC-IBR1 can more easily undergo terminal methylation to yield NOC-IBR2.

## 2. Results and Discussion

### 2.1. Genome Mining

The genome sequence of *Nocardia* sp. CS682 was analyzed with antiSMASH 7.0 to obtain 46 putative BGCs encoding butyrolactones, ectoine, indole, lantibiotics, melanin, nonribosomal peptides, polyketides, siderophores, terpenes, etc. (Appendix A) [20]. The genomic region spanning from 770,703 to 906,587 nt was annotated with type II and type I PKSs, whereas the sequence corresponding to type I PKSs matched the *ngn* BGC. Bioinformatics analysis of the *fnq* gene cluster revealed that the SARP regulator gene *fnqO* is located between two polyketide BGCs via a type I PKS BGC corresponding to *ngn* and an unknown type II PKS BGC. An in-depth analysis of type II PKSs revealed that they matched the putative BGC for the *fnq* gene cluster (Appendix A). Therefore, the genomic region corresponding to 800,427–826,419 nt was predicted to contain the genes responsible for the biosynthesis of *ifnq* in *Nocardia* sp. CS682.

### 2.2. Functional Characterization of the SARP Regulator

The association of transcriptional regulators with secondary metabolites is a well-established phenomenon. Overall, SARP regulators are commonly associated with secondary metabolite biosynthesis; hence, we used conventional gene overexpression, deletion, and complementation experiments for the functional characterization of a regulatory gene, *fnqO.* The production of NOC-IBR1 and NOC-IBR2 in different strains of *Nocardia* is summarized in Figure 1 and Appendix A. In our attempt to evaluate the influence of the SARP regulator (*fnqO*) bordering the *ngn* BGC, recombinant pNV18L2-*fnqO* was introduced into *Nocardia* sp. CS682 and *Nocardia* sp. CS682DR to generate *Nocardia* sp. CS682/*fnqO* and *Nocardia* sp. CS682DR*/fnqO*, respectively. After fermentation, the ethyl acetate extracts of the wild-type and recombinant strains were evaluated by HPLC. The overexpression of *fnqO* in *Nocardia* sp. CS682 led to a reduction in nargenicin production at 265 nm according to HPLC (Figure 1b). However, the overexpression of *fnqO* in *Nocardia* sp. CS682DR led to low production of NOC-IBR1 (0.8 mg/L) and substantial production of NOC-IBR2 (2.8 mg/L) (Figure 1d). To confirm the close association of *fnqO* in this regard, we performed a *fnqO* inactivation experiment using CRISPR-Cas9. The deletion mutant strain was confirmed by PCR amplification and restriction mapping of the amplified fragments (Appendix A). The resulting *Nocardia* sp. CS682DRΔ*fnqO* completely abolished NOC-IBR1 and NOC-IBR2 (Figure 1e). Additionally, the complementation strain *Nocardia* sp. CS682DR/Δ*fnqO*/pNV-SARP restored NOC-IBR2 production (Figure 1f); presumably, NOC-IBR1 was converted to NOC-IBR2. Therefore, *fnqO* is a pathway-specific regulator associated with ifnq production in *Nocardia* species.

### 2.3. Identification of the Furanonaphthoquinone Gene Cluster

Bioinformatics analysis [21] of the *Nocardia* sp. CS682 genome identified the *fnq* BGC, which showed a high degree of nucleotide sequence identity and a similar genetic organization as the previously reported *ifn* gene cluster [6]. It contains a single type II PKS, which is responsible for the biosynthesis of NOC-IBR1 and NOC-IBR2 (Figure 2A). Analogous proteins from both groups were compared, and the results revealed shared homologs, as shown in Appendix A. This finding provides strong evidence that the *fnq* gene cluster is responsible for the biosynthesis of IFQs, including NOC-IBR1 and NOC-IBR2, in *Nocardia* sp. CS682. Therefore, we proposed a putative pathway for *fnq* biosynthesis that utilizes eight molecules of malonyl-CoA processed by Type II PKS and cyclase, resulting in intermediates I and II. Baeyer–Villiger monooxygenases (FnqI) modify polyketide scaffolds (II) synthesized by type II PKS, leading to the creation of product III through a C-C bond cleavage reaction [22,23]. Additionally, various putative enzymes participate in the modification of intermediates III to VIII. Intermediate VIII undergoes further modification into NOC-IBR1 and NOC-IBR2 with the involvement of a methyltransferase (ThnM3) (Figure 2B). However, the FNQ BGC lacked an *O*-methyltransferase (*O*-MT) responsible for the terminal methylation of NOC-IBR1 to generate NOC-IBR2. Hence, we screened several *O*-MTs, such as NarM, ThnM1, ThnM2, and ThnM3, for this purpose. Finally, we observed that only ThnM3 was capable of regiospecific methylation [24] to generate NOC-IBR2, as shown in Figure 3. In a previous study, similar metabolites JBIR-76/77 were isolated from *Streptomyces* sp. RI-77, but their BGCs and pathways were not explored [25].

### 2.4. Fermentation, Isolation, and Analytical Methods

For the isolation and structural elucidation of NOC-IBR1 and NOC-IBR2, fermentation of *Nocardia* sp. CS682DR/*fnqO* was carried out in 8 L of DD media. After multiple rounds of prep-HPLC, pure 6.8 mg of NOC-IBR1 and 22.8 mg of NOC-IBR2 were obtained. Their chemical structures were deduced with the help of mass spectrometry, ^1^H-NMR, ^13^C-NMR, rotating frame Overhauser enhancement spectroscopy (ROESY), correlation spectroscopy (COSY), heteronuclear single quantum correlation (HSQC), and heteronuclear multiple bond correlation (HMBC), as shown in Table 1 and Appendix A. Through these analyses, NOC-IBR1 was identified as 6-methoxy-7,8-dihydroxy-1-methylnaphtho [2,3-c] furan-9-one, and NOC-IBR2 was identified as 6,7-dimethoxy-8-hydroxy-1-methylnaphtho [2,3-c] furan-9-one, a new compound. The structure of NOC-IBR2 is the same as that of NOC-IBR1, but there is an additional methoxy group. For NOC-IBR2, HMBC and ROESY correlations demonstrated that the methoxy group (δH 3.71, δC 60.33) is located at C-7 on the A-ring.

6-Methoxy-7,8-dihydroxy-1-methylnaphtho [2,3-c] furan-9-one (NOC-IBR1) was isolated as a pale yellow solid. The molecular formula of NOC-IBR1 was determined to be C_14_H_12_O_5_ based on HR-ESI-MS analysis, where the HR-QTOF *m*/*z* value was 261.0758 [M + H]^+^ (calculated for C_14_H_13_O_5_, 261.0757).

6,7-Dimethoxy-8-hydroxy-1-methylnaphtho [2,3-c] furan-9-one (NOC-IBR2) was isolated as a pale yellow solid. The molecular formula of NOC-IBR2 was determined to be C_15_H_14_O_5_ based on HR-ESI-MS analysis, where the HR-QTOF *m*/*z* value was 275.0903 [M + H]^+^ (calculated for C_15_H_15_O_5_, 275.0914).

### 2.5. Evaluation of Microbial Inhibition

The antimicrobial activity of NOC-IBR1 and NOC-IBR2 was evaluated using the paper disk diffusion method. NOC-IBR1 exhibited no antimicrobial activity against the tested pathogens. Conversely, NOC-IBR2 showed significant antimicrobial activity against *S. aureus* subsp., as evidenced by the clear inhibition zones (Appendix A).

As summarized in Table 2, NOC-IBR2 exhibited antimicrobial activity against *S. aureus* CCARM 3640, *S. aureus* CCARM 3634, *S. aureus* CCARM 33591, *S. aureus* CCARM 0204, *S. aureus* CCARM 3090, *S. aureus* CCARM 3635, *S. aureus* CCARM 3089, *S. aureus* CCARM 0205, and *S. aureus* CCARM 0027, with MIC values of 7.81, 7.81, 15.63, 31.25, 31.25, 31.25, 62.5, 62.5, and 125 µg/mL, respectively.

To carry out the MBC test, cultured samples with MIC compounds and experimental strains were inoculated on new MHB media (Table 2). For each strain tested, the MBC values were the same as or greater than the MIC values. The MBCs of NOC-IBR2 for the following strains, *S. aureus* CCARM 3640, *S. aureus* CCARM 3634, *S. aureus* CCARM 33591, *S. aureus* CCARM 0204, *S. aureus* CCARM 3090, *S. aureus* CCARM 3635, *S. aureus* CCARM 3089, *S. aureus* CCARM 0205, and *S. aureus* CCARM 0027, were 15.63, 15.63, 31.25, 31.25, 31.25, 62.5, 62.5, 125, and 250 µg/mL, respectively. Thus, the methylated derivative NOC-IBR2 exhibited greater antimicrobial activity than NOC-IBR1.

### 2.6. Evaluation of Anticancer and Antioxidant Properties

NOC-IBR1 and NOC-IBR2 were further subjected to cytotoxicity assays against four different cancer cell lines using a colorimetric test of 3-(4,5-dimethylthiazol-2-yl)-2,5-diphenyltetrazolium bromide (MTT) (Figure 4). The 50% inhibitory concentration (IC_50_) values of NOC-IBR1 and NOC-IBR2 for the A549, Huh7, HeLa, and U87MG cell lines were 199.5, 44.44, 61.14, 119.6, and 112.5, 54.18, 29.27, and 118.5, respectively, but for the normal HaCaT cell line, the IC_50_ value was >200 µM for both compounds. Similarly, the IC_50_ values of doxorubicin in the above cell lines were also calculated (Appendix A). NOC-IBR2 effectively inhibited the viability of A549 lung cancer cells and HeLa cells compared with NOC-IBR1. These findings further indicate that NOC-IBR1 and NOC-IBR2 substantially reduce the viability of A549, Huh7, HeLa, and U87MG cells in a dose-dependent manner. This is the first report on the cytotoxicity of these two compounds against cancer cell lines. At the concentration of 250 μM, ascorbic acid (positive control) and NOC-IBR2 showed similar ABTS radical scavenging activities (Appendix A). It was observed that NOC-IBR2 showed greater antioxidant activity than NOC-IBR1.

### 2.7. Kinetic Behavior of ThnM3 in Response to NOC-IBR1

The steady-state kinetic parameters of ThnM3 were determined with NOC-IBR1. After determining the impact of the enzyme concentration and conducting a time-course analysis, the initial velocity was measured after 15 min using 5 µg of ThnM3 (*O*-MT). The estimated apparent *K_m_* value of NOC-IBR1 with saturating SAM (2 mM) was 15.62 ± 1.75 µM. The *K_m_* value for SAM was estimated to be 39.7 ± 6.08 µM with near-saturating NOC-IBR1 (100 µM) (Figure 5). Appendix A displays the apparent *K_m_* and *V*_max_ values.

## 3. Materials and Methods

### 3.1. Microorganisms and Chemicals

The vector used for cloning was pGEM-T Easy (Promega, Madison, WI, USA). Watertown, Massachusetts, USA-based Addgene provided the pCRISPomyces-2 vector used in this study. Using the *Nocardia–E. coli* shuttle vector pNV18, *Nocardia* sp. CS682 was genetically modified. DNA manipulation was performed using *E. coli* XL1 Blue MRF (Stratagene, La Jolla, CA, USA). *E. coli* ET12567 was used to propagate nonmethylated DNA. Every *E. coli* strain was grown at 37 °C in solid or broth LB media (MB cell, SeoCho-Gu, Seoul, Republic of Korea). Brain heart infusion (BHI) (MB cell, SeoCho-Gu, Seoul, Republic of Korea) served as the initial medium for seeding and regenerating recombinants, and the DD medium was used as the production medium. The following concentrations of antibiotics were used: 100 μg/mL ampicillin, 50 μg/mL apramycin, and 150 μg/mL thiostrepton. Genomic DNA (gDNA) was extracted from *Nocardia* sp. CS682 using the Qiagen DNeasy Tissue Kit (Qiagen, Gaithersburg, MD, USA). gDNA was then used as a template for PCR amplification. Mallinckrodt Baker (Phillipsburg, NJ, USA) provided acetonitrile and trifluoroacetic acid (TFA) of HPLC grade. Additional chemicals were obtained from commercially available sources and were of superior quality. Daejung Chemicals and Metals Co., Ltd. (Shiheung, Republic of Korea) supplied all the remaining organic solvents. Sigma-Aldrich (St. Louis, MO, USA) provided dimethyl sulfoxide-*d*_6_ (DMSO-*d*_6_). Standard molecular biology procedures were followed to isolate, digest, ligate, and manipulate DNA. The recombinant plasmids were verified by DNA sequencing (Macrogen, Inc., Seoul, Republic of Korea), and Appendix A list all the strains, plasmids, and PCR primers used in this study.

### 3.2. Genome Mining

The genomic sequence of *Nocardia* sp. CS682 (GenBank accession no: CP029710.1) was analyzed using antiSMASH 7.0 to annotate 46 putative BGCs (Appendix A) [20]. The genomic regions spanning 770, 703–906, and 587 nt containing type II PKSs and type I PKSs were annotated extensively; the sequence corresponding to type I PKSs matched the *ngn* BGC, and the sequence of type II PKSs matched the putative *fnq* BGC (Appendix A). However, this *fnq* BGC lacks an *O*-methyltransferase (OMT) essential for terminal methylation of NOC-IBR1 to generate NOC-IBR2. Therefore, to support this biotransformation, we attempted to identify a suitable OMT among NarM, ThnM1, ThnM2, and ThnM3.

### 3.3. Genetic Manipulation of the SARP Regulator

The presence of a unique SARP regulator adjacent to the *ngn* BGC spurred our interest in exploring its role in metabolite biosynthesis. For the overexpression of this SARP regulator (FnqO), a codon-optimized gene devoid of the TTA codon (corresponding to the *bldA* tRNA) was synthesized according to the sequence provided in Appendix A and cloned and inserted into the *Bam*HI and *Kpn*I sites of pNV18L2 to generate pNV18L2-*fnqO*. This recombinant strain was then transformed into *Nocardia* sp. CS682 and *Nocardia* sp. CS682DR through electroporation according to a previously published protocol. The selected transformants were grown at 37 °C for 7 days at 200 rpm (SI 600R Richmond Scientific, Chorley, UK), and secondary metabolites were examined via HPLC. To evaluate the impact of the SARP regulator on secondary metabolite production, we generated a SARP regulator deletion mutant in *Nocardia* sp. CS682DR using CRISPR-Cas9. For this purpose, pCRISPomyces-2 was used to clone guide RNA (gRNA) and homologous recombination fragments. The gRNA was synthesized as described in Appendix A and ligated to the *Bbs*I site of pCRISPomyces-2 using a previously described protocol [26,27]. In contrast, homologous fragments were amplified by PCR with specific primers (Appendix A) and cloned and inserted into the *Xba*I site of pCRISPomyces-2 [27]. After the deletion construct was transformed into *Nocardia* sp. CS682DR, the *Nocardia* sp. CS682DRΔ*fnqO* strain was successfully generated. After transformation confirmation, the recombinants were cultured on BHI agar plates supplemented with apramycin for additional generations. Then, colonies that did not grow on BHI agar/apramycin plates were selected, and the homologous recombination strain was identified by double crossover. The legitimate deletion strain *Nocardia* sp. CS682DRΔ*fnqO* was confirmed by PCR and DNA sequencing. The deletion strain was complemented with pNV18L2-*fnqO* to generate *Nocardia* sp. CS682DRΔ*fnqO*/pNV18L2-*fnqO* to evaluate the possibility of polar effects by gene deletion. Metabolites of *Nocardia* sp. CS682DR and its overexpression, deletion, and complementation strains were evaluated using HPLC and LC-HRMS analyses.

### 3.4. In Vitro Assays of O-Methyltransferase

Following expression, purification, and quantification, ThnM3 was used in the enzymatic reaction to validate its catalytic function. The ability of ThnM3 to introduce a methyl group to NOC-IBR1 was assessed in a general reaction that contained 50 μg of ThnM3, 2 mM SAM, 200 μM substrate, and Tris buffer (50 mM, pH 8) in a final volume of 200 μL. Similarly, a control reaction was performed without heat-inactivated ThnM3. The reaction mixtures were incubated at 40 °C for 30 min. To determine the kinetic parameters of NOC-IBR1, reaction mixtures containing ThnM3 (5 μg), SAM (2 mM), Tris buffer (50 mM, pH 8), and various concentrations of NOC-IBR1 (5–200 μM) were prepared in a total volume of 200 μL. Similarly, 200 μL reaction mixtures comprising ThnM3 (5 μg), NOC-IBR1 (100 μM), Tris buffer (50 mM, pH 8), and different concentrations of SAM (5–200 mM) were prepared to determine the kinetic parameters of ThnM3 for SAM. Different enzyme concentrations for ThnM3 assays were tested for 30 min in a 200 μL reaction mixture containing ThnM3 (1–5 μg), 5 μM NOC-IBR1, 2 mM SAM, and Tris buffer (50 mM, pH 8). ThnM3 (5 μg), 2 mM SAM, 5 μM NOC-IBR1, and Tris buffer (50 mM, pH 8.0) were used to measure the time-dependent in vitro conversion of NOC-IBR1 to NOC-IBR2 every 5 min for 30 min. To stop each reaction, 200 μL (double volume) of chilled methanol was added, and HPLC was used to examine the supernatant after 30 min of centrifugation at 10,000× *g*. Data were obtained in triplicate, and appropriate control reactions were maintained. The data are reported as the mean and standard deviation. Nonlinear regression analysis was used to calculate the values of *Vmax* and the Michaelis–Menten constant, *K_m_*, under the assumption of steady-state Michaelis–Menten kinetics [28].

### 3.5. Fermentation and Extraction

Five hundred-milliliter flasks containing 50 mL of BHI medium were used to cultivate *Nocardia* sp. CS682DR as well as the overexpression, deletion, and complementation strains. Following a two-day incubation at 37 °C, 20 mL of seed culture was added to each of the 5 L fermenters (XP-50 Fermentec, Republic of Korea), which contained 3 L of DD medium (0.2% oatmeal, 0.2% yeast extract, 0.2% soybean meal, 0.1% calcium chloride, 1% maltose, 0.1% magnesium chloride, and 0.4% glycerol) for fermentation in a bioreactor. The fermenters were shaken at 400 rpm and 37 °C. After 5 days, the resulting culture was mixed with a double volume of ethyl acetate and shaken for 5–7 h. A rotary evaporator was used to dry the extracts, and methanol was added. The concentrated sample was filtered with an HPLC Syringe Filter (Catalog; 10463060, Whatman, UK). Subsequently, 20 μL of the filtered extracts were analyzed using reverse-phase HPLC-PDA coupled with a C18 column (Mightysil RP-18 GP (4.6 × 250 mm, 5 μM) at 320 nm) to identify peaks corresponding to various secondary metabolites, including NOC-IBR1/2. Binary conditions were used with solvent 1 (water containing 0.05% trifluoroacetic acid buffer) and solvent 2 (100% acetonitrile (ACN)), maintaining a flow rate of 1 mL/min for 30 min; gradient conditions included 10% (0–2 min), 70% (20–24 min), 100% (24–28 min), and 50% (28–30 min) ACN. HR-qTOF ESI/MS analysis was performed with SYNAPT G2-S (Waters Corp., Milford, MA, USA) in a positive ion mode. The target compounds (NOC-IBR1/2) were purified by prep-HPLC using a C18 column (YMC-Pack ODS-AQ-HG, 250 × 20 mm, 10 µM, Shimogyo-Ku, Kyoto, Japan) coupled to a UV detector (320 nm) and a 40 min binary gradient at a flow rate of 10 mL/min with 1.2 mL injection volume to the loop volume of 2 mL. The sample was filtered with an HPLC Syringe Filter (Catalog; 10463060, Whatman, UK). The concentration of ACN in the binary gradient was as follows: 0–10 min, 0–15%; 15–20 min, 50%; 20–25 min, 80%; 25–24 min, 95%; 34–39 min, 10%; and finally stopped at 40 min with a flow rate of 10%. The peaks for NOC-IBR1 and NOC-IBR2 corresponding to retention times (*t*_R_) of 26 min and 29 min in prep-HPLC with a minimal UV threshold of 1000 U, respectively, were collected. The collected fractions from prep-HPLC were reanalyzed using analytical HPLC to determine their purity. Following lyophilization, the purified products were dissolved in deuterated DMSO-*d*_6_ obtained from Sigma-Aldrich and subjected to nuclear magnetic resonance (NMR) spectroscopy (700 MHz) to determine their structures. For structural confirmation, two-dimensional nuclear magnetic resonance (NMR) techniques such as heteronuclear single-quantum correlation (HSQC) and heteronuclear multiple bond correlation (HMBC) were used in addition to ^1^H-NMR and ^13^C-NMR.

### 3.6. Antimicrobial Assays

Antimicrobial assays of NOC-IBR1 and NOC-IBR2 were carried out against a panel of eighteen different pathogens. Among them, fourteen were Gram-positive bacteria (MRSA) (*Staphylococcus. aureus* CCARM 3640, *S. aureus* CCARM 3089, *S. aureus* CCARM 33591, *S. aureus* CCARM 0205, *S. aureus* CCARM 0204, *S. aureus* CCARM 0027, *S. aureus* CCARM 3090, *S. aureus* CCARM 3634, *S. aureus* CCARM 3635, *Bacillus subtilis* ATCC 6633, *Enterococcus faecalis* 19433, *Enterococcus faecalis* 19434, *Kocuria rhizophilla* NBRC 12708, and *Micrococcus luteus*), and four were Gram-negative bacteria (*Escherichia coli* ATCC 25922, *Proteus hauseri* NBRC 3851, *Klebsiella pneumonia* ATCC10031, and *Salmonella enterica* ATCC 14028). The paper disk diffusion assays on Mueller–Hinton agar plates (MHA) were performed according to the Kirby–Bauer method and Clinical Laboratory Standard Institute (CLSI) guidelines [29,30]. NOC-IBR1 and NOC-IBR2 were applied to paper disks to adjust the final concentration to 40 µg/disk. MHA plates were inoculated with 1 × 10^8^ colony-forming units (CFUs)/mL of bacterial culture. Using sterile paper plates measuring 6 mm (Advantec, Toyo Roshi Kaisha, Ltd., Tokyo, Japan), a 40 µg/disk compound was applied to the surface of the infected agar plates. The samples were then incubated for 18–20 h at 37 °C. The inhibition diameter zone was then measured, calculated for each pathogen, and expressed in millimeters (mm). Since all the compounds were dissolved in dimethyl sulfoxide (DMSO), it was used as a solvent control, and erythromycin was used as a positive control.

The minimum inhibitory concentration (MIC) and the minimum bactericidal concentration (MBC) of NOC-IBR1 and NOC-IBR2 were estimated using a previously described method [31]. The broth dilution method was then used to calculate the MICs of NOC-IBR1, NOC-IBR2, and erythromycin as positive controls against nine strains of the Gram-positive bacterium *S. aureus* subsp. Mueller–Hinton broth (MHB, Difco, Baltimore, MD, USA), and the assays were performed in duplicate in 96-well plates. The nine MRSA strains were subsequently grown in MHB and subjected to MIC and MBC assays. To determine the MIC, the broth dilution method was applied [32]. In a 96-well plate, MHB and the samples were dispensed and diluted serially. In all the wells except for the negative control (broth only), the bacterial cell suspension was adjusted to 0.5 McFarland standard (1 × 10^8^ CFU/mL) and diluted to 2.5 × 10^6^ CFU/mL in MHB. The cells were then cultured for 16–20 h at 37 °C. The MBC assay was carried out on an MHB medium by inoculating cultured samples containing MIC compounds for 18 h at 37 °C, and the MICs of the bacterial strains were determined. The assays were repeated twice to maximize reliability and reproducibility.

### 3.7. Anticancer and Antioxidant Assays

Lung cancer cells (A549), liver cancer cells (Huh7), cervical cancer cells (HeLa), brain cancer cells (U87MG), and a normal cell line (human keratinocyte cell line (HaCaT)) were acquired from the Korean Cell Line Bank (Seoul, Republic of Korea). Ten percent fetal bovine serum (FBS) was added to Dulbecco’s modified Eagle’s medium (DMEM) to culture the HeLa A549 cells. Similarly, Huh7 cells were grown in DMEM obtained from Corning Cellgro Manassas, VA, USA. Glioblastoma U87MG cells were cultured in a minimum essential medium (MEM) supplemented with 10% FBS. DMEM supplemented with 10% FBS, 100 µg/mL streptomycin, and 100 µg/mL benzylpenicillin was used to grow HaCaT cell lines. Each cell line was kept in an incubator with 5% CO_2_ that was humidified at 37 °C (Thermo Scientific, Vantaa, Finland). For the cell growth assays, various cancer cells at 2 × 10^3^ cells/well were plated in 96-well culture plates. NOC-IBR1 and NOC-IBR2 were added to each well at various concentrations (200 μM, 100 μM, 50 μM, 25 μM, 12.5 μM, 6.25 μM, 3.16 μM, 1.56 μM, and 0.78 μM), and the cells were incubated for 72 h. A 3-(4,5-dimethylthiazol-2-yl)-2,5-diphenyltetrazolium bromide (MTT) colorimetric assay was used to measure cell growth (cell viability).

The ABTS radical cation decolorization assay was performed to evaluate the free radical scavenging potential of NOC-IBR1 and NOC-IBR2 using ascorbic acid as a standard reference [33,34]. The decrease in absorbance signifies the ability of the samples to neutralize or scavenge the ABTS^+^· radical, indicating their free radical scavenging capacity. The ability of NOC-IBR1 and NOC-IBR2 to eliminate free radicals was evaluated using the ABTS radical cation decolorization test [35]. ABTS (7 mM) in deionized water and 2.45 mM potassium persulfate (1:1) were combined to form the ABTS^+^· cation radical, which was then kept at room temperature in the dark for 12 to 16 h before use. After that, ethanol or deionized water was added to the ABTS^+•^ solution to dilute it until the absorbance at 734 nm was 0.70. In 96-well plates, 10 μL of the sample was combined with 195 μL of ABTS^+•^ solution, and in the blank well, 195 μL of ABTS^+^ solution was combined with 10 μL of DMSO and incubated at room temperature for 30 min. Then, the absorbance was measured at 734 nm. Ascorbic acid was used as a standard reference, and all determinations were performed in triplicate.

### 3.8. Statistical Analysis

The findings are presented as the means ± standard errors (SEs). Statistical significance between the two groups, comprising the control and test groups, was assessed using the Student’s *t*-test. A *p*-value less than 0.05 was considered indicative of statistical significance.

## 4. Conclusions

Using the integrated approach of genome mining and genome editing, we identified and characterized two novel quinoline compounds, NOC-IBR1 and NOC-IBR2, from *Nocardia* sp. CS682 DR/fnqO. Furthermore, a biocatalysis approach was utilized for mining and characterizing the post-modification enzyme ThnM3, which is involved in methylation reactions. The kinetic parameters of this enzyme were also determined in this study. Furthermore, NOC-IBR2 displayed notably stronger antimicrobial activity than did NOC-IBR1 against a range of human pathogens, including various drug-resistant strains of *S. aureus*. Additionally, the evaluation of the antioxidant and anticancer properties of NOC-IBR2 showed promising potential. Thus, this study supports the importance of an integrated approach involving genome mining, genome editing, and biocatalysis to explore new specialized metabolites in diverse microorganisms, including rare actinobacteria such as *Nocardia.*

## Figures and Tables

**Figure 1 ijms-25-08847-f001:**
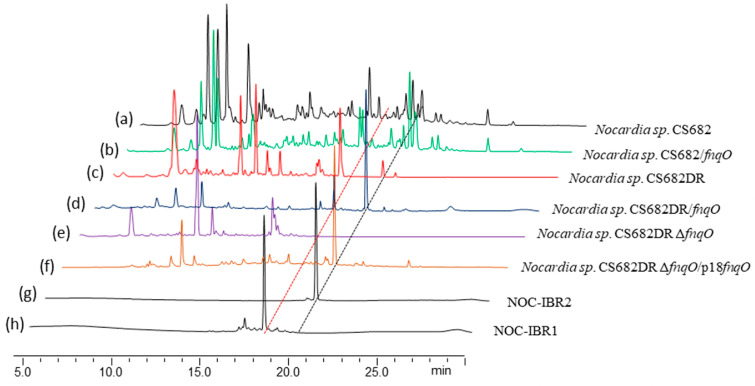
HPLC chromatograms of extracts from different strains: (**a**) wild-type *Nocardia* sp. CS682; (**b**) *Nocardia* sp. CS682/*fnqO* overexpression strain; (**c**) PKS deletion mutant *Nocardia* sp. CS682DR; (**d**) *Nocardia* sp. CS682DR/*fnqO* overexpression strain; (**e**) deletion mutant *Nocardia* sp. CS682DRΔfnqO; (**f**) complementation strain *Nocardia* sp. CS682DRΔf*nqO*/pNV18L2-*fnqO*; (**g**) purified compound NOC-IBR2 from *Nocardia* sp. CS682DR *fnqO* overexpression strain; and (**h**) purified compound NOC-IBR1 from the *Nocardia* sp. CS682DR *fnqO* overexpression strain.

**Figure 2 ijms-25-08847-f002:**
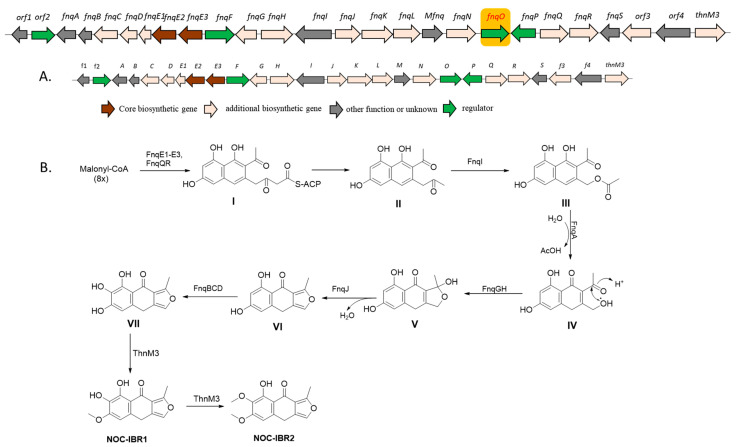
(**A**) Organization of the NOC-IBR1 and NOC-IBR2 biosynthesis-related gene clusters from *Nocardia* sp. CS682DR; the translational directions of the genes are indicated by arrows, and the letters above the arrows represent the gene names. (**B**) Proposed biosynthetic pathway for compounds NOC-IBR1 and NOC-IBR2, where FnqE1/E2/E3 are acyl carrier proteins (ACPs), FnqI is a Bayer–Villiger monooxygenase, and FnqA is an esterase. FnqGH = SnoaL-like domain protein and FAD oxidase; FnqJ = hydrolase. FnqBCD = ferredoxin, cytochrome P450, and flavin reductase, respectively; ThnM3 = methyltransferase. *fnqO* in the orange frame is the SARP gene.

**Figure 3 ijms-25-08847-f003:**
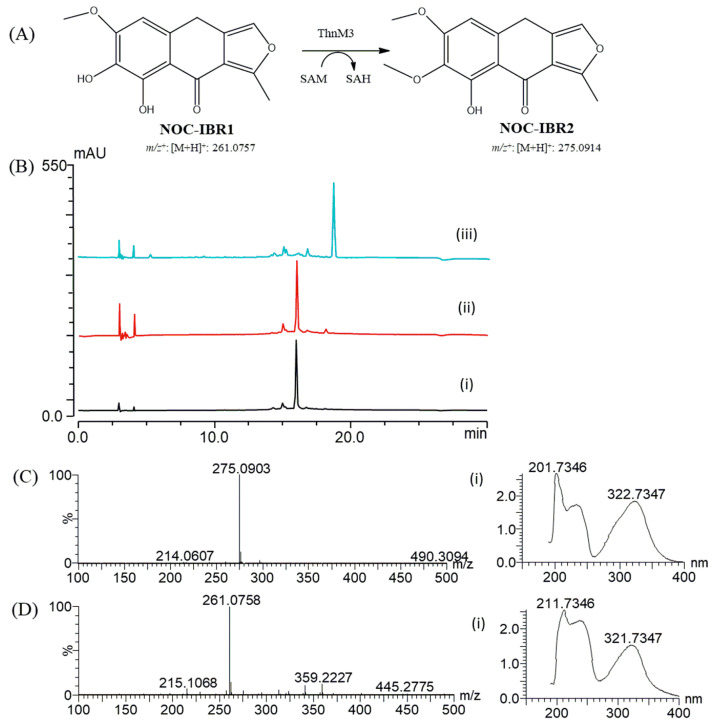
(**A**) Methylation of NOC-IBR1 by ThnM3 in the presence of SAM at 40 °C for 3 h (50 mM Tris HCl buffer (pH 8), 2 mM SAM, 2 mM substrate, and 50 µg of ThnM3 protein). (**B**) HPLC-PDA chromatogram analyses of (**i**) purified NOC-IBR1, (**ii**) the NOC-IBR1 enzyme with NarM, and (**iii**) the NOC-IBR1 enzyme with ThnM3 in the presence of SAM at 40 °C. (**C**) HQ-QTOF ESI/MS spectrum of the peak corresponding to NOC-IBR2 showing an *m*/*z* value of 275.0903 (error: 5 ppm). (**C**) (**i**) UV/VIS spectrum of NOC-IBR2. (**D**) HQ-QTOF ESI/MS spectrum of the peak corresponding to NOC-IBR1 showing an *m*/*z* value of 261.0758 (error: 0.03 ppm). (**D**) (**i**) UV/VIS of NOC-IBR1.

**Figure 4 ijms-25-08847-f004:**
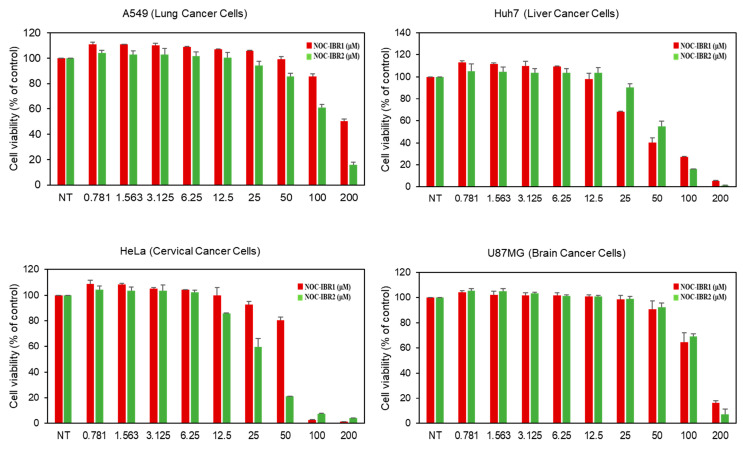
The cytotoxicity of NOC-IBR1 and NOC-IBR2 was assessed across different cell lines. Each cell line was maintained in a 5% CO_2_ humidified incubator at 37 °C. The cells were seeded at a density of 2 × 10^3^ cells/well in 96-well culture plates for the cell growth assay. Various concentrations of the compounds (200 μM, 100 μM, 50 μM, 25 μM, 12.5 μM, 6.25 μM, 3.16 μM, 1.56 μM, and 0.78 μM) were added to each well, and the cells were incubated for 72 h. Cell growth was measured using the MTT colorimetric assay.

**Figure 5 ijms-25-08847-f005:**
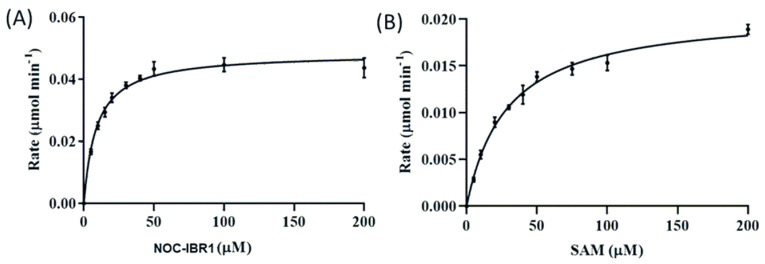
Enzyme-kinetic characteristics of ThnM3. (**A**) Kinetic analysis using various concentrations of NOC−IBR1 (5–200 µM), ThnM3 (5 µg), SAM (2 mM), and Tris-HCl buffer (50 mM, pH 7.5). (**B**) Kinetic analysis using various concentrations of SAM (5–200 µM), ThnM3 (5 µg), NOC−IBR1 (100 µM), and Tris-HCl buffer (50 mM, pH 7.5).

**Table 1 ijms-25-08847-t001:** Comparison of ^1^H- and ^13^C-NMR chemical shifts of NOC-IBR1 and NOC-IBR2 measured in DMSO-d6.

	^1^H-NMR (700 MHz, DMSO-*d6*)		^13^C-NMR (176 MHz, DMSO-*d6*)	
Position	NOC-IBR1	NOC-IBR2	NOC-IBR1	NOC-IBR2
1			157.40	157.68
2				
3	7.64 (*s*,1H)	7.66 (t, *J* = 2.0 Hz, 1H)	136.96	137.08
3a			121.58	121.31
4	4.04 (*s*,2H)	4.08 (*s*,2H)	23.87	24.26
4a			133.78	139.21
5	6.65 (*s*,1H)	6.70 (*s*,1H)	104.41	104.45
6			153.74	158.36
6-O-CH_3_	3.87 (*s*,3H)	3.89 (*s*,3H)	56.33	56.50
7			132.19	134.25
7-OH	8.54 (*s*,1H)			
7-O-CH_3_		3.71 (*s*,3H)		60.33
8			152.03	157.19
8-OH	13.11 (*s*,1H)	13.29 (*s*,1H)		
8a			111.83	112.14
9			187.52	187.41
9a			116.23	116.01
10	2.67 (*s*,3H)	2.67 (*s*,3H)	14.31	14.32

**Table 2 ijms-25-08847-t002:** MIC and MBC of NOC-IBR2 and positive control erythromycin (Erm) against nine strains of MRSA.

	MIC (µg/mL)		MBC (µg/mL)	
MDR Strains	NOC-IBR2	Erm	NOC-IBR2	Erm
*S. aureus* CCARM 3640	7.81	>1000	15.63	>1000
*S. aureus* CCARM 3634	7.81	>1000	15.63	>1000
*S. aureus* CCARM 33591	15.63	>1000	31.25	>1000
*S. aureus* CCARM 0204	31.25	3.90	31.25	7.81
*S. aureus* CCARM 3090	31.25	500	31.25	500
*S. aureus* CCARM 3635	31.25	500	62.50	500
*S. aureus* CCARM 3089	62.50	>1000	62.50	>1000
*S. aureus* CCARM 0205	62.50	3.90	125	7.81
*S. aureus* CCARM 0027	125	3.90	250	7.81

## Data Availability

The data used in this research are mentioned in the article and Appendix A.

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
