# Peer review of "Genome Mining and Genetic Manipulation Reveal New Isofuranonaphthoquinones in Nocardia Species"

_ijms, 2024, doi:10.3390/ijms25168847_

Round 1
Reviewer 1 Report
Comments and Suggestions for Authors
The manuscript (ID ijms-3144892) submitted for review contains important new information regarding the use of new biologically active substances from actinomycetes (Nocardia). I agree with the authors that the identification of specialized metabolites isolated from microorganisms is urgently needed to determine their role in cancer treatment and control of multidrug-resistant pathogens. Especially when new biologically active substances are less toxic to humans during therapy.
The manuscript is well and calmly written. The results are well documented and aptly developed and described.
My minor suggestions for the manuscript:
1) please adapt the work to the journal's requirements, including the way of writing references in the text, Figs.
2) please provide the precision of the devices, e.g. incubation temperature
3) please add information about the manufacturers of reagents/media
Author Response
Response to Reviewer 1
Dear Reviewer,
We appreciate your rigorous, careful, and thoughtful review. Your thoroughness and constructive feedback have been invaluable to us.
We have carefully revised the manuscript, considering all the comments and suggestions provided. The sections that have changed are highlighted in the track change option. We believe this revision has significantly improved our manuscript, making it better suited for publication.
Comments of Reviewer 1
The manuscript (ID ijms-3144892) submitted for review contains important new information regarding the use of new biologically active substances from actinomycetes (Nocardia). I agree with the authors that the identification of specialized metabolites isolated from microorganisms is urgently needed to determine their role in cancer treatment and control of multidrug-resistant pathogens. Especially when new biologically active substances are less toxic to humans during therapy.
The manuscript is well and calmly written. The results are well documented and aptly developed and described.
My minor suggestions for the manuscript:
1) please adapt the work to the journal's requirements, including the way of writing references in the text, Figs.
Response: Thank you, the appropriate correction has been duly made.
2) please provide the precision of the devices, e.g. incubation temperature
Response: Thank you, the appropriate correction has been duly made.
3) please add information about the manufacturers of reagents/media
Response: Thank you for your suggestion. The appropriate correction has been duly made.

Reviewer 2 Report
Comments and Suggestions for Authors
Dear the Editor
Pudel PB et al reported an identification of novel isofuranonahthoquinones in Nocardia species. These authors performed an HPLC-based idenfification of target compound and Nordica species (Figure 1) followed by its identification by NMR (Table 1). Putative methylation pathway was confirmed biochemically with a purified ThnM3 protein (Fig. 3). Cytotoxic effect of these newly identified compounds (NOC-IBR1/2) was further assessed by several human cell lines (Fig. 4). This study was well-designed with solid outcome. Method section was described with proper manner.
Major concerns:
1) Please provide the unit for NOC-IBR concentrations in Fig. 4.
2) In Fig. 3 C/D legend, please describe them more properly.
Author Response
Response to Reviewer 2
Dear Reviewer,
We appreciate your rigorous, careful, and thoughtful review. Your thoroughness and constructive feedback have been invaluable to us.
We have carefully revised the manuscript, considering all the comments and suggestions provided. The sections that have changed are highlighted in the track change option. We believe this revision has significantly improved our manuscript, making it better suited for publication.
Comments of Reviewer 2
Poudel PB et al reported an identification of novel isofuranonahthoquinones in Nocardia species. These authors performed an HPLC-based identification of target compound and Nordica species (Figure 1) followed by its identification by NMR (Table 1). The putative methylation pathway was confirmed biochemically with a purified ThnM3 protein (Fig. 3). Cytotoxic effect of these newly identified compounds (NOC-IBR1/2) was further assessed by several human cell lines (Fig. 4). This study was well-designed with solid outcome. The method section was described in a proper manner.
Major concerns:
1) Please provide the unit for NOC-IBR concentrations in Fig. 4.
Response: Thank you for your suggestion. We have made the appropriate corrections in Figure 4.
2) In Fig. 3 C/D legend, please describe them more properly.
Response: As per your suggestion, the legend section was described more properly.

Reviewer 3 Report
Comments and Suggestions for Authors
The following points needs to be addressed by the authors
1. The author utilized HPLC chromatography for metabolite characterization but did not describe the procedure in detail to allow other researchers to replicate the experimental methodology.
2. Information regarding prefiltration/filtration prior to HPLC is necessary. Was prefiltration conducted to eliminate any solid particles (chunks) from the extracts? Was a guard column utilized before the LC column?
3. The selection of a 320 nm wavelength requires clarification. It is essential to understand why this particular wavelength was chosen and its relevance to the metabolite structure. For instance, why were wavelengths such as 280 nm, 310 nm, or 360 nm not selected? The rationale should be correlated directly with the structure of the metabolite in question.
4. More clarity about how the authors determined the peaks in HPLC. Adequate details on the peak assignment in HPLC are required.
4. The methodology for Prep-HPLC should be elaborated descriptively. Certain details are absent, such as whether there was a threshold UV absorption for fraction collection. If so, this should be reported. Additionally, specify the loop volume, the injection volume, the preparation method, and how the method was developed.
5. Explanation about how the authors identified the peak near 20 minutes as the principal compound peak. Were any standards used in the study? This needs to be clearly stated as it is currently unclear in the manuscript.
6. The NMR chemical shifts must be organized according to the numbering of the structure, therefore a figure is needed where the structure of the compound is numbered alongside the assigned chemical shifts, which will improve the readability of the paper.
7. A correction measurement (in ppm) in HRMS is required. Adherence to the guidelines is necessary as the formula to calculate the deviation is missing.
Please calculate the deviation accordingly.
Author Response
Response to Reviewer 3
Dear Reviewer,
We appreciate your rigorous, careful, and thoughtful review. Your thoroughness and constructive feedback have been invaluable to us.
We have carefully revised the manuscript, considering all the comments and suggestions provided. The sections that have changed are highlighted in the track change option. We believe this revision has significantly improved our manuscript, making it better suited for publication.
Comments of Reviewer 3
The following points needs to be addressed by the authors
- The author utilized HPLC chromatography for metabolite characterization but did not describe the procedure in detail to allow other researchers to replicate the experimental methodology.
Response: The appropriate correction has been duly made.
- Information regarding pre-filtration/filtration prior to HPLC is necessary. Was prefiltration conducted to eliminate any solid particles (chunks) from the extracts? Was a guard column utilized before the LC column?
Response: We filtered the sample with an HPLC Syringe Filter (Catalog; 10463060, Whatman, UK) every time before operating HPLC. Yes, prefiltration was conducted to eliminate any solid particles.
- The selection of a 320 nm wavelength requires clarification. It is essential to understand why this particular wavelength was chosen and its relevance to the metabolite structure. For instance, why were wavelengths such as 280 nm, 310 nm, or 360 nm not selected? The rationale should be correlated directly with the structure of the metabolite in question.
Response: We also did HPLC in different UV regions like 250 nm, 280 nm, 300 nm, 320, and 350 nm but the UV absorbance of the compounds was highest at 320 as indicated in Figure 3, so 320 nm was used for this analysis.
- More clarity about how the authors determined the peaks in HPLC. Adequate details on the peak assignment in HPLC are required.
Response: After overexpression of SARP genes, we saw a new peak in Nocardia sp. CS682DR extract, so we did further mass analysis and NMR study of the two new peaks (Figure 1D) and got two new compounds.
- The methodology for Prep-HPLC should be elaborated descriptively. Certain details are absent, such as whether there was a threshold UV absorption for fraction collection. If so, this should be reported. Additionally, specify the loop volume, the injection volume, the preparation method, and how the method was developed.
Response: The appropriate correction has been duly made.
- Explanation about how the authors identified the peak near 20 minutes as the principal compound peak. Were any standards used in the study? This needs to be clearly stated as it is currently unclear in the manuscript.
Response: Technically, numerous peaks were collected and among them, few were further analyzed by NMR study. Based on those results, the genome mining approach predicts the biosynthetic gene cluster. So, the finding related to NOC-IBR1/2 was summarized in this manuscript
- The NMR chemical shifts must be organized according to the numbering of the structure, therefore a figure is needed where the structure of the compound is numbered alongside the assigned chemical shifts, which will improve the readability of the paper.
Response: Thank you for your suggestion. The structure and numbering of the compound were given in the supplementary figures S3 and S4.
- A correction measurement (in ppm) in HRMS is required. Adherence to the guidelines is necessary as the formula to calculate the deviation is missing.
Please calculate the deviation accordingly.
Response: We have calculated the error ppm as you suggested and found them as mentioned below.
For NOC-IBR1, the error (ppm) in LC-HRMS is 0.03 ppm
For NOC-IBR2, the error (ppm) in LC-HRMS is 5.08 ppm
Based on various literature, this range below or around 5 ppm is acceptable in LC-HRMS analysis.

Reviewer 4 Report
Comments and Suggestions for Authors
The manuscript investigates the use of genome mining and genetic manipulation fro dug discovery from Nocardia Species. Overall, the work is well-designed and well-executed. After carefully reading this work, I have some observations:
1. I suggest rereading the text for typos and grammatical mistakes.
2. The aim of the study is not clearly mentioned in the abstract.
3. The introduction should end with the aims of the study rather than the findings. Rewrite and relocate these parts.
4. Line 337: Review the references.
5. Lines 383-385: DMSO was used as a solvent control. What was used as a positive control for the assay?
6. Line 387: review the citation.
7. Lines 418 and 421: review the citations.
8. Add a schematic diagram to summarize the experimental design.
Comments on the Quality of English LanguageThe work would benefit from close editing.
Author Response
Response to Reviewer 4
Dear Reviewer,
We appreciate your rigorous, careful, and thoughtful review. Your thoroughness and constructive feedback have been invaluable to us.
We have carefully revised the manuscript, considering all the comments and suggestions provided. The sections that have changed are highlighted in the track change option. We believe this revision has significantly improved our manuscript, making it better suited for publication.
Comments of reviewer 4.
The manuscript investigates the use of genome mining and genetic manipulation fro dug discovery from Nocardia Species. Overall, the work is well-designed and well-executed. After carefully reading this work, I have some observations:
- I suggest rereading the text for typos and grammatical mistakes.
Response: Thank you for your suggestion. The appropriate correction has been duly made.
- The aim of the study is not clearly mentioned in the abstract.
Response: The appropriate correction has been duly made in the abstract section.
- The introduction should end with the aims of the study rather than the findings. Rewrite and relocate these parts.
Response: Thank you. The appropriate correction has been duly made in the introduction section.
- Line 337: Review the references.
Response: Thank you. The references have been reviewed and added too.
- Lines 383-385: DMSO was used as a solvent control. What was used as a positive control for the assay?
Response: Erythromycin (Erm) was used as a positive control and mentioned in Table 2 and lines number 399.
- Line 387: review the citation.
Response: Thank you. The references have been removed and added new one.
- Lines 418 and 421: review the citations.
Response: Thank you. The references have been reviewed.
- Add a schematic diagram to summarize the experimental design.
Response: Thank you for your suggestion. The following graphical abstract has been included.

Round 2
Reviewer 2 Report
Comments and Suggestions for Authors
Dear the Editor
All raised concerns were adequately corrected.
Reviewer 4 Report
Comments and Suggestions for Authors
The authors have made substantial improvements and addressed all the suggested comments.